# Value of Cellular Components and Focal Dedifferentiation to Predict the Risk of Metastasis in a Benign-Appearing Extra-Meningeal Solitary Fibrous Tumor: An Original Series from a Tertiary Sarcoma Center

**DOI:** 10.3390/cancers15051441

**Published:** 2023-02-24

**Authors:** Mohammad Hassani, Sungmi Jung, Elaheh Ghodsi, Leila Seddigh, Paul Kooner, Ahmed Aoude, Robert Turcotte

**Affiliations:** 1Division of Orthopedic Surgery, McGill University Health Centre, Montreal, QC H3G 1A4, Canada; 2Department of Orthopedic Surgery, Mashhad University of Medical Sciences, Mashhad 91886-17871, Iran; 3Department of Pathology, McGill University Health Centre, Montreal, QC H4A 3J1, Canada; 4School of Public Health, University of Montreal, Montreal, QC H3N 1X9, Canada; 5Department of Community Medicine, Tehran University of Medical Sciences, Tehran 14117-13135, Iran

**Keywords:** solitary fibrous tumor, extrameningeal, dedifferentiation, cellular variant

## Abstract

**Simple Summary:**

A solitary fibrous tumor (SFT) is a fibroblastic mesenchymal tumor with the hallmark of an NAB2–STAT6 gene fusion and an intermediate tendency to metastasize. Based on the lack of a histologic-based grading system for extra-meningeal SFTs, we defined the prognostic value of histologic features that predict the risk of developing distant metastases. Moreover, our study revealed the histologic alterations to recurrent SFTs that affect the biological behavior of the tumor.

**Abstract:**

Histology has not been accepted as a valid predictor of the biological behavior of extra-meningeal solitary fibrous tumors (SFTs). Based on the lack of a histologic grading system, a risk stratification model is accepted by the WHO to predict the risk of metastasis; however, the model shows some limitations to predict the aggressive behavior of a low-risk/benign-appearing tumor. We conducted a retrospective study based on medical records of 51 primary extra-meningeal SFT patients treated surgically with a median follow-up of 60 months. Tumor size (*p* = 0.001), mitotic activity (*p* = 0.003), and cellular variants (*p* = 0.001) were statistically associated with the development of distant metastases. In cox regression analysis for metastasis outcome, a one-centimeter increment in tumor size enhanced the expected metastasis hazard by 21% during the follow-up time (HR = 1.21, CI 95% (1.08–1.35)), and each increase in the number of mitotic figures escalated the expected hazard of metastasis by 20% (HR = 1.2, CI 95% (1.06–1.34)). Recurrent SFTs presented with higher mitotic activity and increased the likelihood of distant metastasis (*p* = 0.003, HR = 12.68, CI 95% (2.31–69.5)). All SFTs with focal dedifferentiation developed metastases during follow-up. Our findings also revealed that assembling risk models based on a diagnostic biopsy underestimated the probability of developing metastasis in extra-meningeal SFTs.

## 1. Introduction

Solitary fibrous tumors (SFTs) are a subset of fibroblastic mesenchymal soft tissue tumors with the hallmark of an NAB2–STAT6 gene fusion [1,2,3] and present a metastatic rate of up to 34% after surgical resection [4,5]. The evolution of SFT histopathology has abolished some previous tumor entities, such as hemangiopericytoma; however, histology has not been accepted as a valid predictor of SFT biological behavior [2,6,7]. Many studies have tried to determine some criteria to differentiate benign and malignant variants of the tumor. Some suggested histologic features, including mitosis, atypia, and tumor necrosis, as well as demographic features, including age, tumor size, and location of the tumor as predictors of SFT aggressive behavior. Particularly size and mitotic counts have been emphasized more than others [5,8,9,10,11,12]. Based on the lack of a histologic grading system, Demicco et al. developed a risk stratification model based on the patient’s age, tumor size, mitotic count, and tumor necrosis. The model is accepted by the World Health Organization (WHO) to predict the risk of metastasis in extra-meningeal SFT patients [7,13,14,15]. Not only a histologically benign-appearing SFT, but a low-risk SFT according to Demicco’s model may present with distant metastases [11,16]. Furthermore, the patient’s age, which is comparable to histologic items in Demicco’s model, has inconsistent predictive value for the biological behavior of the tumor [4,11,16,17]. A histology-based grading system for SFTs originating from the central nervous system (CNS) is available whilst debating on the risk assessment of extra-meningeal SFTs [18,19]. The current study aimed to contribute to the debates on the biological behavior of extra-meningeal SFTs and to find some characteristics that predict the risk of metastases in a low-risk SFT.

## 2. Methods

### 2.1. Patients

Upon receiving approval from the Research Ethics Board (REB), we collected the medical records of 99 patients who were diagnosed with one of the SFT-related pathology reports, which included solitary fibrous tumor, hemangiopericytoma, anaplastic hemangiopericytoma, and malignant solitary fibrous tumor. The inclusion criterion was any patients with a primary extra-meningeal solitary fibrous tumor that was treated with surgical resection and followed within our tertiary sarcoma center. The diagnosis of SFT in our cases was based on histological features supported by ancillary immunohistochemical tests, such as STAT6, CD34, CD99, Vimentin, and BCL2. Any questionable diagnoses were reassessed by a sarcoma pathologist (S.J). The exclusion criteria were 1, SFT originating from the meningeal membrane or central nervous system (CNS), 2, metastatic lesions without a primary tumor histologic report, and 3, patients lacking appropriate follow-up data. We excluded 16 patients who had a tumor originating from the central nervous system (CNS); furthermore, 32 patients were excluded from the final statistical analysis because of having incomplete medical records or being diagnosed with metastasis at initial presentation without primary tumor pathology available. Ultimately, 51 primary extra-meningeal SFTs that were treated with surgical resection between 2005 to 2021 were included in the final analysis. The histologic characteristics of each tumor, such as frequency of mitosis per 10 high power fields (HPFs), a total necrotic area equal to or more than 10% of the tumor, having a hypercellular component, having a focal dedifferentiated nodule, and cyto-atypia, were extracted from the pathology reports of the first diagnostic biopsy and the resected tumor.

### 2.2. Statistical Analysis

Categorical variables included tumor necrosis, atypia, cellularity, the patient’s gender, and the tumor’s primary location. Numerical variables included the patient’s age, tumor size, and mitosis count. Simultaneously, the age was stratified into <55 and ≥55 to be assessed as a categorical variable. Categorical variables were presented by frequency and valid percent (VP), and continuous variables were presented by median, range, and standard deviation (SD). The metastatic/recurrence risk score was calculated according to Demicco’s [14] and Pasquali’s [9] models for each patient. The en bloc resection of the primary tumor was considered index surgery. The outcomes were distant metastasis, which we considered as an indicator of aggressive behavior of the tumor, and the patient’s five-year disease-free (5YDF) survival after the index surgery. As we excluded the patients with incomplete follow-up from our final analysis, we used the logistic regression model, Fisher exact test, and Chi-square test to address any statistical association between variables and distant metastases, using SPSS version 25 and Stata version 14. We also performed cox regression analysis and applied Kaplan–Meier survival curve analysis to estimate the probability of outcomes during the follow-up. Receiver operating characteristic (ROC) curve analysis was used to determine the optimal values with the highest sensitivity and specificity for each continuous variable correlated with outcomes.

## 3. Results

### 3.1. Patients

Nine patients (18%) had distant metastasis in the follow-up period after the index surgery. Seven patients (14%) presented with local recurrence, and 78% of our patients have not shown local or distant tumor recurrence during the first 5-year surveillance after the index surgery. No simultaneous metastatic lesion had been found at the time of the index surgery; however, only one case was suspicious as having lung metastasis at the time of the tumor diagnosis. Table 1 presents the patient demographic and histology data.

### 3.2. Distant Metastasis

Distant metastases were detected in nine (18%) of our patients. The median time to distant metastasis was 24 months (range: 6–72) (Figure 1).

Gender and age had no significant statistical association with SFT distant metastasis (*p* values were 0.47 and 0.20, respectively). No significance was reached by dichotomizing age into younger than 55 and older than 55 (*p* = 0.82) when looking at metastasis. SFT location had no significant association with distant metastasis (*p* = 0.42). Nonetheless, only one intra-thoracic tumor, which had a focal dedifferentiated component, presented with metastasis. Tumor size increased the risk of distant metastasis (*p* = 0.001). In cox regression analysis for metastasis outcome, a one-centimeter increment in tumor size enhanced the expected metastasis hazard by 21% during the follow-up time (HR = 1.21, CI 95% (1.08–1.35)). ROC curve analysis showed that the optimal cut-point for tumor size to distinguish metastatic from non-metastatic outcomes is 7.4 cm (area under the curve (AUC) = 0.85).

Our analysis demonstrated that the cellular variant of the tumor (Figure 2) was associated with distant metastasis (*p* = 0.001). The Kaplan–Meier cure showed considerable differences between the two groups regarding the presence of a cellular component (Figure 3). Atypia (*p* = 0.12) and necrosis (*p* = 0.073) had no significant association with metastasis in our patients. The mitotic count was correlated with distant metastasis (*p* = 0.003). During follow-up, each increase in the number of mitotic figures escalated the expected hazard of metastasis by 20% (HR = 1.2, CI 95% (1.06–1.34)). Our analysis showed that a threshold of 1.5 mitoses was associated with an increased metastatic rate, using ROC curve analysis (AUC = 0.82).

Seven patients developed local recurrence. The mean mitotic counts per 10 HPFs in the primary and the recurrent lesions were 4.7 and 13.8, respectively. Tumor local recurrence was associated with a higher probability of developing distant metastasis in our patients (*p* = 0.003) (HR = 12.68, CI 95% (2.31–69.5)) (Figure 4).

### 3.3. Patient Survival

Thirty-one (78%) patients had no local recurrence or distant metastases during the 5-year follow-up after the index surgery, of which twelve had a post-index surgery surveillance of less than 60 months (Figure 5).

In this study, the patient’s age (*p* = 0.12) and gender (*p* = 0.58), tumor location (*p* = 0.56, 0.93 for different layers), cellular variant (*p* = 0.41), atypia (*p* = 0.54), and necrosis (*p* = 0.98) had no impact on the patient’s 5YDF survival. Even the mitotic count had no statistical association with the patient’s 5YDF survival (*p* = 0.38).

In the multivariable cox regression model, only the tumor size decreased the likelihood of having 5YDF survival in our patients (*p* = 0.01) (HR = 0.79, CI 95% (0.67–0.94)). Table 2 presents the statistical association between variables and outcomes.

### 3.4. Risk Assessment Tools

The cumulative risk score based on Demicco’s model was linked to a higher chance of developing distant metastasis (*p* = 0.007, HR = 1.62) and decreased the likelihood of having 5YDF survival (*p* = 0.019, HR = 1.63), using the pathologic report after the primary tumor resection. Categorical Demicco’s score based on risk stratification groups also had a statistical relationship with metastasis and patient 5YDF survival (*p* values were 0.01, 0.004 respectively). Nevertheless, two patients from the low-risk group presented with distant metastases, and three patients from the high-risk group had no metastases during their follow-up (range 13–120 months).

In extra-plural SFTs, categorical Pasquali’s risk score based on the histology of the index surgery was associated with distant metastasis (*p* = 0.051); however, it did not show statistical significance based on 5YDF survival (*p* = 0.063).

The risk score calculation based on diagnostic biopsy histology failed to predict the likelihood of distant metastasis in our patients, using both Demicco’s (*p* = 0.81) and Pasquali’s (*p* = 0.15) models.

### 3.5. Dedifferentiated SFT

Dedifferentiation in our cases was defined by a high-grade sarcoma in an abrupt transition from typical SFT (Figure 6). A focal dedifferentiated component was detected in four primary SFTs (7.8%), which were resected with negative surgical margins. All these patients presented with distant metastasis during the surveillance after the resection (range 21–48 months). Two tumors had a cellular component present in their diagnostic biopsy samples (Table 3).

## 4. Discussion

Histology grading has been accepted as a principal predictor of metastatic behavior in many soft tissue sarcomas [20,21]; however, there is no consensus on a histologic-based grading system for extra-meningeal SFT, which encompasses a spectrum of tumors from dedifferentiated variants to benign-appearing SFTs. Moreover, Demicco’s risk stratification model has been accepted as a reliable statistical prognostic measure by the WHO, but it shows some limitations in predicting the aggressive behavior of the tumor [16]. Similarly, a few patients in our study, being classified as low risk due to Demicco’s model, experienced distant metastases.

Age is a prognostic factor in the Demicco model; however, our study found that age as either a numerical or categorical variable had no significant association with the risk of developing metastases. On the contrary, hypercellularity as a histologic feature was reported to increase the odds of developing metastasis in previous studies, of which, some had the drawback of including meningeal SFTs [9,10,22,23]. Comparably, our study revealed that the cellular variant of the tumor in extra-meningeal SFT increased the risk of distant metastasis. Having no definite cut-point, cellularity is considered a subjective finding by critics [5]. Furthermore, it has not been considered a prognostic factor in the FNCLCC grading system. Our findings, however, strongly suggest that cellularity increases the prognostic value of risk assessment models much more than age, which will be aligned with the meningeal/CNS SFT grading system [18,19].

The current study revealed that we will underestimate the metastatic potential of an SFT if we assemble a risk model based on core needle biopsy information. To overcome this limitation, we must consider any prognostic factor that independently impacts the biological behavior of an SFT, to devise our treatment plan.

Tumor size was reported as a prognostic factor by many studies, and different cut-point values ranging from 8 to 15 cm have been calculated [8,17,24]. According to our findings, a tumor size larger than 7 cm, which is associated with higher odds of metastasis and independently impacts the patient’s survival, should be approached as a tumor with a high probability of aggressive behavior.

Mitotic count was also associated with metastasis and impacted patient survival, either as an independent factor or as a part of risk assessment models in many studies [4,5,8,9]. Our findings, similarly, supported the prognostic value of mitotic activity (≥2 mitosis/10 HPFs) in extra meningeal SFTs.

A solitary fibrous tumor can present with a synchronous focal dedifferentiated area [25,26]. In our patients, any primary tumor with a focal dedifferentiated nodule was associated with distant metastasis, albeit being surrounded by a benign-appearing SFT, and developing metastasis was independent of their negative surgical margins. Half of our patients with focal dedifferentiation were reported as having a cellular variant of the tumor in their diagnostic biopsy. Similar findings were revealed by other studies, which highlighted the aggressive behavior of focal dedifferentiated tumors, as well as the association between hypercellularity and the focal dedifferentiation of SFTs [23,27].

Our recurrent tumors showed more aggressive histologic features, such as higher mitotic rates. We also found a significant association between the risk of developing distant metastasis and local recurrence, albeit not being an independent variable. These important findings favored the aggressive treatment approach to an SFT with any histologic or demographic worrisome features; however, we need further studies to determine independent predictors for the above-mentioned finding.

Regarding the flourishing role of molecular studies in the diagnosis of soft tissue sarcomas [7,28,29], one study replaced the age with the MIB-1 proliferation index in their risk assessment model to predict the outcomes of SFT patients [30]. We assume that the molecular study overcomes the current flaws of predicting the biological behavior of an extra-meningeal SFT.

Our retrospective study had some inherent limitations: we deliberately excluded any patient with incomplete medical records, which might increase the risk of selection bias in our study. Considering the low frequency of outcomes in our five-year follow-up period, the results of cox regression analysis should be interpreted with caution; furthermore, it restricted the use of multivariate analysis to determine independent associations between variables and outcomes. In addition, half of our patients were followed for less than 5 years and may present with a metastatic lesion in the future considering the risk of late metastasis with SFTs [16,17]. Finally, we excluded molecular findings from our variables due to the lack of a complete molecular profile for all patients.

## 5. Conclusions

Hypercellular components and focal dedifferentiation predict the risk of developing metastases in extra-meningeal SFTs. Assembling a risk model based on biopsy features underestimates the metastatic potential of the tumor. Recurrent SFTs present with aggressive histologic features, such as higher mitotic activity, and increase the likelihood of distant metastasis.

## Figures and Tables

**Figure 1 cancers-15-01441-f001:**
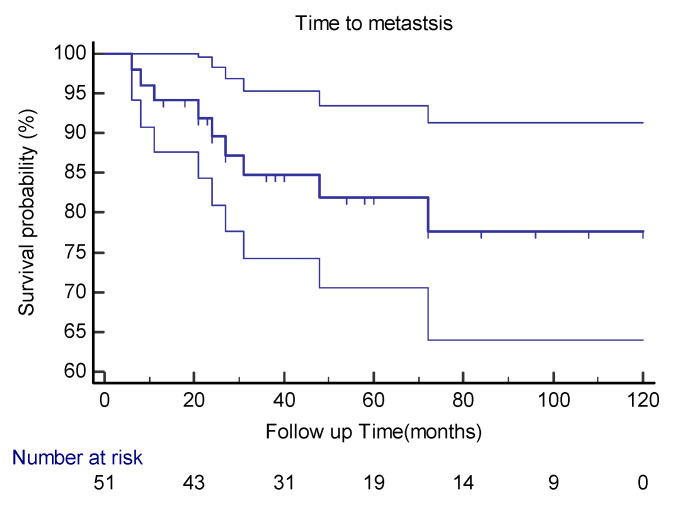
Kaplan–Meier curve estimates with 95%CIs for distant metastasis in extra-meningeal SFT patients.

**Figure 2 cancers-15-01441-f002:**
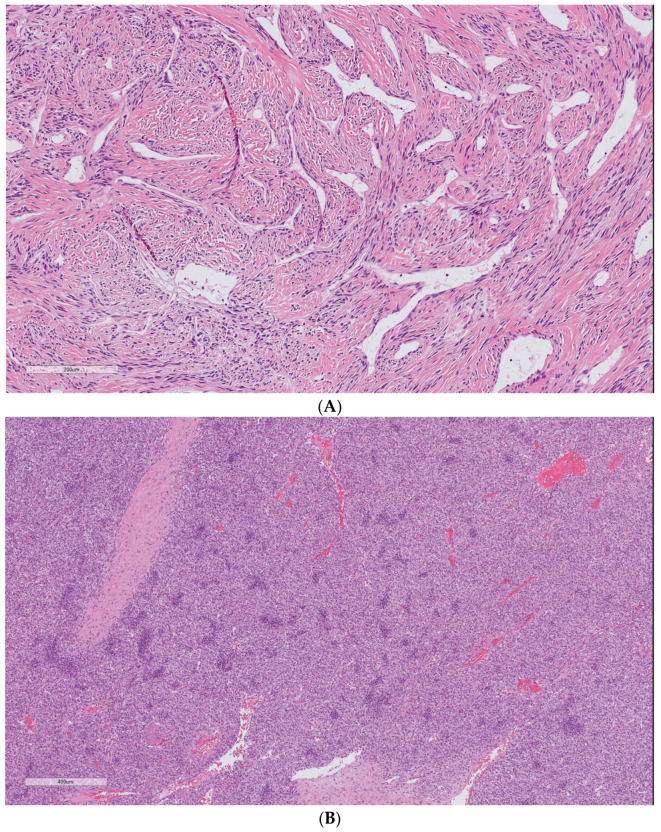
Solitary fibrous tumor H&E staining. The spectrum of cells and fibrous stroma in: (**A**) classic hyalinized solitary fibrous tumor and (**B**) cellular variant of the tumor. Cellular SFT is characterized by the tightly packed proliferation of ovoid to spindle cells arranged around conspicuous vessels and scant stromal components. The uncropped high-quality slides are shown in Appendix A.

**Figure 3 cancers-15-01441-f003:**
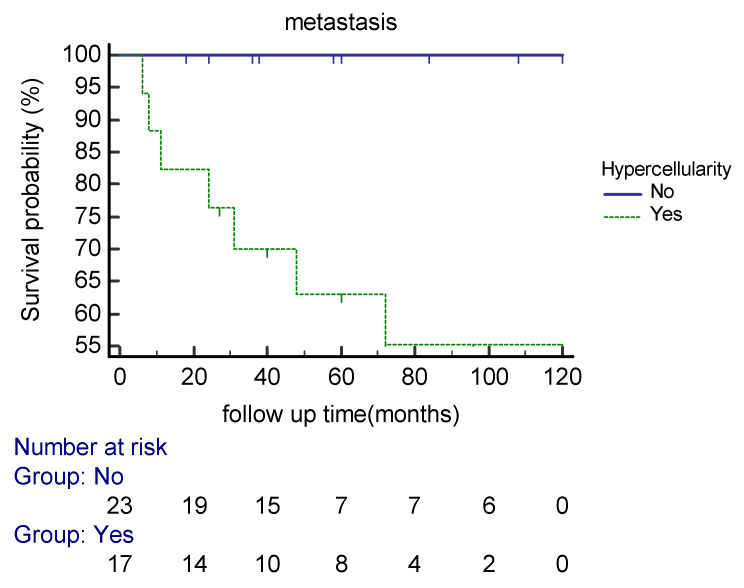
Kaplan–Meier curve estimates of the probability of distant metastasis in extra-meningeal SFT patients based on hypercellularity during the follow-up period.

**Figure 4 cancers-15-01441-f004:**
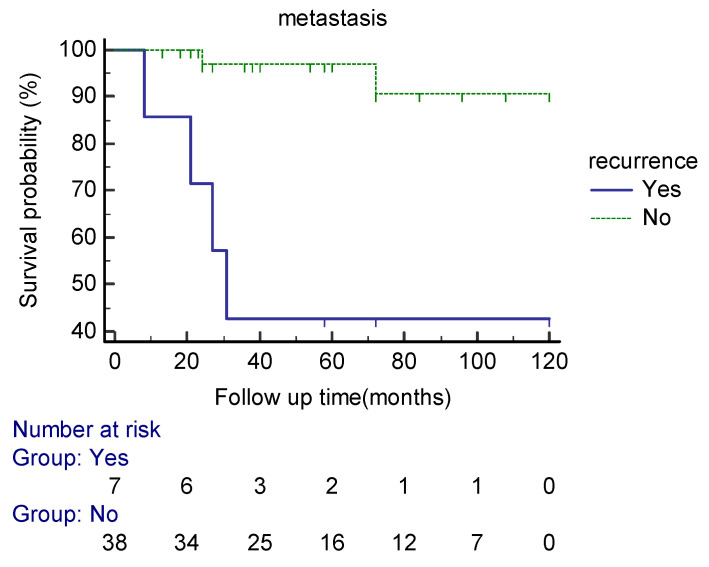
Kaplan–Meier curve estimates of the probability of distant metastasis in extra-meningeal SFT patients based on the local recurrence status during the follow up period.

**Figure 5 cancers-15-01441-f005:**
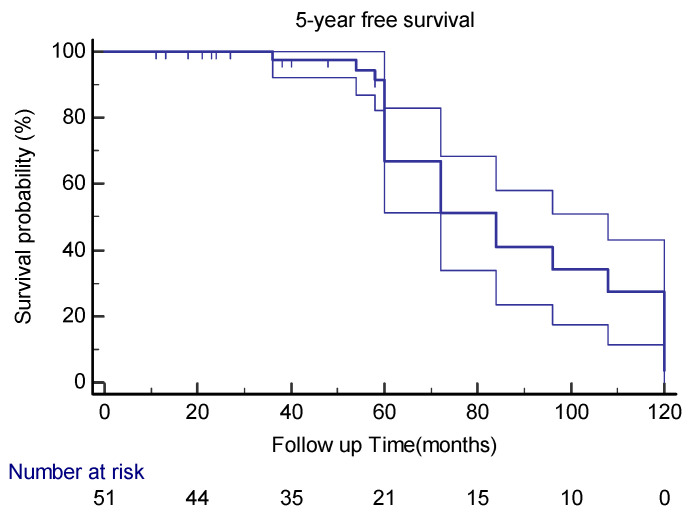
Kaplan–Meier curve estimates with 95%CIs for 5-year disease-free survival in extra-meningeal SFT patients.

**Figure 6 cancers-15-01441-f006:**
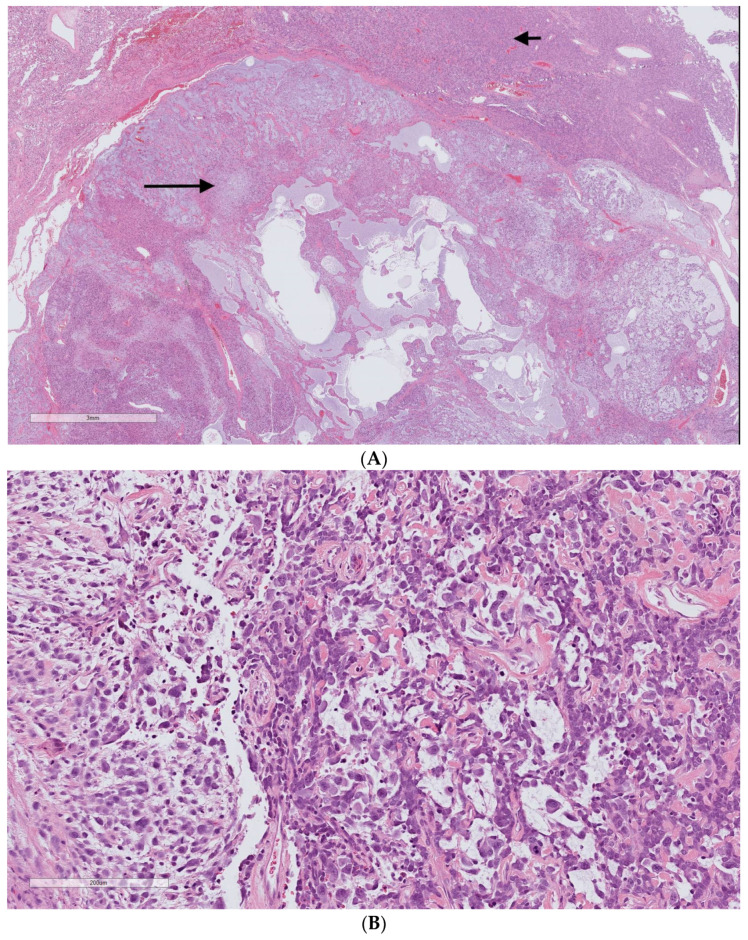
Dedifferentiated solitary fibrous tumor H&E staining. (**A**) Two distinct areas of the typical solitary fibrous tumor (upper part/short arrow) and high-grade pleomorphic sarcoma (lower part/long arrow). (**B**) High-grade undifferentiated sarcoma. The uncropped high-quality slides are shown in Appendix A.

**Table 1 cancers-15-01441-t001:** Patient demographic and histology data. VP: valid percent, SD: standard deviation, HPF: high power field, N: number, m: months.

Variables	Median (Range)	N (VP%)	Mean (SD)
Age (Y)	59 (27–81)		
Gender			
male		27 (53%)	
female		24 (47%)	
Tumor location			
Extremity/intramuscular		20 (39%)	
Retroperitoneum/viscera		18 (35%)	
Intra-thoracic		13 (25%)	
Follow-up (m)	60 (11–120)		
Tumor size (cm)	5.5 (1.1–23)		7.61 (5.03)
Necrosis		11 (23.4%)	
Cellularity		17 (42.5%)	
Atypia		18 (39.1%)	
Mitosis (10HPF)	2.5 (0–22)		4.15 (4.9)
Demicco’s risk score			
low		32 (62.7%)	
Intermediate		13 (25.5%)	
High		12 (32.4%)	
Pasquali’s risk score (extra pleural SFT)			
Very low/low		27 (72.9)	
Intermediate		5 (13.5)	
High		12 (32.4)	

**Table 2 cancers-15-01441-t002:** The statistical association between variables and outcomes are presented; HR: hazard ratio, CI: confidence interval, 5YDF: 5-year disease-free.

Variables	Metastasis	HR (CI 95%)	5YDF Survival	HR (CI 95%)
Age (continuous)	*p* = 0.20		*p* = 0.12	
Age (categorical)	*p* = 0.82		*p* = 0.45	
Hypercellularity	*p* = 0.001		*p* = 0.41	
Size	*p* = 0.001	1.21 (1.08–1.35)	*p* = 0.01	0.79 (0.67–0.94)
Mitotic activity	*p* = 0.003	1.2 (1.06–1.34)	*p* = 0.38	
Demicco score	*p* = 0.007	1.62 (1.14–2.3)	*p* = 0.019	1.63 (1.08–2.45)
Local recurrence	*p* = 0.003	12.68 (2.31–69.5)		

**Table 3 cancers-15-01441-t003:** Characteristics of extra-meningeal SFT patients with a dedifferentiated component. ND: not determined. Int: intermediate. HPF: high power field.

Case	Age	Location	Size cm	Mitosis 10HPF	Hyper Cellular	Demicco Risk Group	Dedifferentiated Nodule Histology
1	31	Paraspinal muscle	7.5	2	Yes	Low	Scattered epithelioid/rhabdoid change, negative CD34
2	59	Pre-sacral	14	10	ND	Int.	Focal high-grade dedifferentiated sarcoma, negative CD34
3	56	Pleura	11	13	Yes	Int.	Focal rhabdoid change
4	41	Paraspinal muscle	8.5	5	ND	Low	Focal myxoid/epithelioid change, negative CD34

## Data Availability

Upon reasonable request, the study data and analyses are available from the first author.

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
