# Peer review of "Value of Cellular Components and Focal Dedifferentiation to Predict the Risk of Metastasis in a Benign-Appearing Extra-Meningeal Solitary Fibrous Tumor: An Original Series from a Tertiary Sarcoma Center"

_cancers, 2023, doi:10.3390/cancers15051441_

Round 1
Reviewer 1 Report
This is an interesting study owing to the rather large number of patients (n=51) with extra-meningeal SFTs. However, the results are unfortunately obscured by methodological shortfalls.
Authors should better describe how does their study relate to risk-stratification scores previously published, in particular by Demicco (2012).
Indeed, they state that they address "benign-looking" extra-meningeal SFTs but do not clearly define this population (it does not overlap with the "low-risk" group from Demicco's score).
It would have been interesting to clearly describe in which groups are the patient according to each risk-stratification method (Demicco, Pasquali, present study), using a summary plot for instance. Paragraph 3.4 is not very informative in this regard.
Moreover, figures are not easy to follow and important information are missing, eg: censored values and number of patients in KM curve, numbers or dots rather than bars in histograms, table 1 is missing headlines, etc.
Statistically, we would have expected a log-rank test to assess differences between groups and a Cox model with hazard ratios to analyze association between disease free survival and prognostic factors. it does not seem to be the case here (?)
In the end, the two main prognostic factors emphasized in the title, cellular component and focal dedifferenciation seem to apply to only 7 (?) and 4 patients with metastasis respectively (which are probably not enough to state that they "predict the risk of metastasis" as in the title)
Overall, the study should benefit from methodological help to increase the scientific robustness of their claims.
Author Response
Dear Reviewer,
We would like to express our gratitude for your taking the time to review our manuscript; we appreciate it. We tried to answer all the questions and revise our manuscript and figures based on your insightful comments; nonetheless, we had some limitations in this retrospective study.
Q1: Authors should better describe how does their study relate to risk-stratification scores previously published, in particular by Demicco (2012), Indeed, they state that they address "benign-looking" extra-meningeal SFTs but do not clearly define this population (it does not overlap with the "low-risk" group from Demicco's score).
A1: The previous studies showed that either the histologically benign-looking SFTs or low-risk groups based on Demicco’s model may present aggressive behavior. To address your comments, and to differentiate those entities, we revised the introduction, methods, and results (sections 1, 2.1, 3.4).
Q2: It would have been interesting to clearly describe in which groups are the patient according to each risk-stratification method (Demicco, Pasquali, present study), using a summary plot for instance. Paragraph 3.4 is not very informative in this regard.
A2: The goal of section 3.4 was to determine if risk models could not predict the aggressive behavior of some of our low-risk cases. Two low-risk cases based on Demicco’s model, and two low-risk cases based on Pasquali’s model presented with metastases. We revised the section to clarify the message.
Q3: Moreover, figures are not easy to follow and important information are missing, eg: censored values and number of patients in KM curve, numbers or dots rather than bars in histograms, table 1 is missing headlines
A3: We revised the figures and tables based on the reviewers’ comments. Figure one related to metastasis has been changed and consists of censored and number of patients. Figure two replaced with Kaplan-Meier curve estimates for metastasis based on hypercellularity.
Q4: Statistically, we would have expected a log-rank test to assess differences between groups and a Cox model with hazard ratios to analyze association between disease free survival and prognostic factors. it does not seem to be the case here
A4: According to comments, we did cox regression analysis for the survival outcomes. The manuscript and figures revised based on new analysis.
Q5: In the end, the two main prognostic factors emphasized in the title, cellular component and focal dedifferentiation seem to apply to only 7 (?) and 4 patients with metastasis respectively (which are probably not enough to state that they "predict the risk of metastasis" as in the title)
A5: Nine patients (18%) developed metastasis during surveillance, of which seven had a documented cellular component. Four patients had a dedifferentiated nodule, and all presented with metastases. Accordingly, the study findings strongly support the predictive value of cellularity and dedifferentiation.

Reviewer 2 Report
Reviewer comments are attached as a PDF.

Author Response
Dear Reviewer,
We would like to thank you for the thoughtful and comprehensive review of our manuscript. We tried to respond to all the questions and revise our manuscript and figures based on your valuable comments; however, we had some limitations in this retrospective study.
Kindly Find our point-by-point response attached.
Sincerely,

Reviewer 3 Report
This manuscript entitled “The Value of Cellular Component and Focal Dedifferentiation to Predict the Risk of Metastasis in a Benign Looking Extra-Meningeal Solitary Fibrous Tumor: An original Series from a Tertiary Sarcoma Center.” describes the detection of the factors which could predict the distant metastasis of extra-meningeal solitary fibrous tumors. They proposed the tumor size, mitotic count and existence of dedifferentiation might correlate to the distant metastasis and disease-free survival. The cohort might have the enough strength, and the parameters which they surveyed looks appropriate, nevertheless, the description has many immature points as well as the lack of important description. The manuscript should be entirely revised before publication. The points to be addressed is as follows:
Major point
1. The authors should clarify the reason why they analysed extra-meningeal SFT, that the biological characteristics might be same among meningeal and extra-meningeal SFT.
2. Currently, the histologic grading system has been established in WHO blue book 5th edition at least regarding meningeal SFT, nevertheless, authors did not refer to this grading system in the manuscript. Authors should designate the research in regard to that criterion. At least authors should clarify why they designate the threshold of mitotic count at 10/10HPF nevertheless WHO histological grade designated the threshold at 5/10HPF.
3. Size of the tumor might be also associated with the time course (stage) of the tumor, nevertheless authors emphasize that the tumor size should be a parameter of histological grading. It seems like the failure of logic. Authors should therefore treat this parameter distinctively from histological parameters.
4. In the histological grading system in WHO blue book employed necrosis as an important factor which can predict the prognoses. How was the correlation of necrosis to the tumor metastasis in this study? Authors should address this issue in the revised manuscript.
5. Solitary fibrous tumor contains many histological variations. Authors should manifest the definition of the histological diagnostic criteria in the methods part. Also, authors clarify whether genetic (detection of STAT6-NAB rearrangement) or immunohistochemial manifestation of STAT6 had been executed or not.
6. The way of presentation seems very immature. For example, regarding Figure 1, authors should divide the group into cellular and non-cellular, then present the population of metastatic and non-metastatic.
7. The results shown in Fig. 4 should be presented in Caplan-Meyer curve as well as to be conjugated with the result shown in Fig. 1.
7. Authors described that dedifferentiated component could be the important point, nevertheless, the description seems lacked in the definition of dedifferentiated SFT component. Authors should indicate the definition and append the histological figures.
8. How was the correlation between MIB-1 labeling index and prognosis?
9. Authors should revise the conclusions more precisely.
Minor points:
1. In Figure 3, markings which show significant differences are required.
2. Authors should cite the newest WHO blue books in the reference.
Author Response
Dear Reviewer,
We sincerely appreciate your time and insightful comments. We tried to answer explicitly the questions and to revise our manuscript and figures based on your expert opinion; nevertheless, we had some limitations in this retrospective study.
Kindly Find our point-by-point responses attached.
Sincerely,

Reviewer 4 Report
There are some minor comments.
It would be better to describe how authors could diagnose 51 cases of SFT (for example, immunostaining for STAT6).
It would be better to define "hypercellular component" criteria in this study.
It would be better to add gross and histopathologic figures.
Regarding four cases of dedifferentiated SFTs, adding histologic findings of dedifferentiated areas would be better.
Please check Table 1.
Age (Y) Median 59 Range 27-81
Demico's score -> Demicco's score
Author Response
Dear Reviewer,
We are extremely grateful for having your expert feedback on our manuscript; we appreciate it. Although we had some limitations in this retrospective study, we tried our best to answer all the questions and revised our manuscript and figures based on your insightful comments.
R1: It would be better to describe how authors could diagnose 51 cases of SFT (for example, immunostaining for STAT6).
A1: We revised the methods (2.2 section) to explain how we reached the diagnosis.
R2: It would be better to define "hypercellular component" criteria in this study.
A2: As we mentioned in the discussion, the hypercellular variant of the tumor is mostly a subjective parameter and no well-defined cut-off has been determined, and we found some degree of interobserver discrepancy to evaluate cellularity [1,2]. In our study, the hypercellular variant and a hypercellular component were interchangeably utilized in comparison to the hypocellular hyalinized variant.
R3: It would be better to add gross and histopathologic figures.
A3: Having the histological photos was not part of our proposal, but we can consider it in case it is necessary to explain the study findings.
R4: Regarding four cases of dedifferentiated SFTs, adding histologic findings of dedifferentiated areas would be better.
A3,4: Pathologic description of the dedifferentiated nodules was added to the manuscript.
R5: Please check Table 1.
A5: Table 1 was revised and corrected.
- Yamada, Y.; Kohashi, K.; Kinoshita, I.; Yamamoto, H.; Iwasaki, T.; Yoshimoto, M.; Ishihara, S.; Toda, Y.; Ito, Y.; Kuma, Y.; et al. Histological background of dedifferentiated solitary fibrous tumour. J Clin Pathol 2022, 75, 397-403, doi:10.1136/jclinpath-2020-207311.
- Salas, S.; Resseguier, N.; Blay, J.-Y.; Le Cesne, A.; Italiano, A.; Chevreau, C.; Rosset, P.; Isambert, N.; Soulie, P.; Cupissol, D. Prediction of local and metastatic recurrence in solitary fibrous tumor: construction of a risk calculator in a multicenter cohort from the French Sarcoma Group (FSG) database. Annals of oncology 2017, 28, 1779-1787.
Kindly find our point-by-point responses attached.
Sincerely,

Round 2
Author Response
Reviewer 2, Round 2
February 3, 2023
Dear Reviewer,
We would like to express our gratitude for your taking the time to review our manuscript; we appreciate it. We tried to answer all the questions and revise our manuscript and figures based on your insightful comments; nonetheless, we had some limitations in this retrospective study.
- The analysis is incomplete/not comprehensive.
a. Although the authors mentioned in the limitations that they do not include any
molecular analysis due to a lack of complete molecular profile of the patients, this is still a weakness of the manuscript. To fully understand which tumor properties contribute to the aggressive nature of the cancer, analyzing what molecular markers are available is desirable (such as STAT6, NAB2-STAT6 fusion breakpoint, etc).
Authors answer: Aa: The diagnosis of SFT in our cases was based on histological features supported by ancillary immunohistochemical tests such as STAT6, CD34, CD99, Vimentin, and BCL2. Any questionable diagnoses were reassessed by a sarcoma pathologist (S.J). The importance of STAT6 as a highly sensitive immunohistochemical marker for SFT has been shown in the literature since 2013; nonetheless, our retrospective studies do not have STAT6 for all their cases (especially cases of which the index had been done before 2015)
Reviewers’ response: No change was made, and it doesn’t sound like there were great pathological or genetic tests done on the original patient dataset.
Answer: A) The histologic diagnosis of SFT is relatively straightforward. Any questionable cases were confirmed by Immuno-stains for STAT6 which is a surrogate protein of the NAB2-STAT6 fusions. Furthermore, some cases of these studies had been diagnosed before the time that the importance of STAT6 was shown in the literature.
- B) The NAB2-STAT6 fusions define the SFT, but the STAT6/or the status of NAB2-STAT6 plays no significant role in risk/prognostic assessment.
Having STAT 6 profile for all patients was not feasible in our retrospective study.
- The table of patients’ demographic and histology data is not comprehensive and poorly formatted. Since there are only 51 patients analyzed. A table listing each patient and features such as age, sex, anatomical site, tumor size (cm), surgical procedure, surgical margins, necrosis, cellularity, atypia, mitosis, DeMicco Score, Local Recurrence, Follow- up (months), status, etc. If you want to group the patients together, at least format the table correctly and include overall survival (OS).
Authors answer: Ab: Table 1 was adjusted and reformatted based on the reviewers’ comments.
Reviewers’ response: Although change was made to include tumor location, the table still doesn’t include all the patients in the study. The table still doesn’t include follow-up information like OS, status, and recurrence rates.
Answer: The recurrence rate and metastases rate were mentioned in section 3.1 (line 106 and 107). Most SFT cases have indolent biological behavior, and the 5-y-OS in the previous studies was about 86% (1). The 5-y- disease-free-survival in our series was 78% (line 107); however, the OS was not among our objectives.
- Paper needs proper formatting and technical editing.
a. Figures 3 and 4 are never referenced in the main text.
Authors response: A2a: Regarding reviewer comments, we have changed all graphs with better explanation and resolution; besides, the figures were referenced in the manuscript.
Reviewers’ response: Fixed.
- The figure of speech is confusing and unclear at times and needs to be rephased. For example, “one-centimeter increase in tumor size increased the odds of metastasis by 28%.” Increase in size as compared to what? 1 cm increase in size over a set time (say 1 month). For each 1 cm increase in size of the tumor the probability of metastases is increased by 28%? Please clarify. A line-graph figure may help make this point more clear.
Authors response: A2b: According to comments, we did cox regression analysis for metastasis outcome. Since the effect size is the hazard ratio and the predictor is a quantitative variable, we interpreted it as one unit increase. Therefore, we changed the representing sentences in sections 3.2 and 3.3
Reviewers’ response: Thank you for modifying the sentence for clarification. However, the statement is still ambiguous. Now the median follow-up is 60 months. So, are you implying there is a 21% chance of metastasis if the tumor grew by 1-cm or more over 60 months? Or are you implying there is a 21% chance of metastasis if the tumor grew by 1-cm at the patients next surveillance scan (typically 3 months to a year)?
Answer: For further explanation, the interpretation has been changed to:” In cox regression analysis for metastasis outcome, there is a 21% increase in the expected metastasis hazard relative to a one-centimeter increase in tumor size during the follow-up time (HR=1.21, CI 95% (1.08-1.35)). The expected metastasis hazard is 21% higher in a tumor which is one-centimeter larger than other ones. The correction was added to the manuscript (line 123-125), and we hope that the modification will resolve the ambiguity.
- The paper is written in past and present tense. The authors need to pick one and stick to it.
Authors response: A c-e: Major revision was performed.
Reviewers’ response: Better. Still a mix of tenses, but less.
The manuscript has been revised to address your concern.
- In results section 3.1. You need to capitalize the first word of the sentence, “patient’s demographic and...” Moreover, revise the manuscript for professional writing. For example, the sentence reads better as, "Table 1 presents the patients' demographic and histological data."
Authors response: A c-e: Major revision was performed.
Reviewers’ response: Better.
- Too many semicolons. For clarity, the authors need to revise the manuscript to break these statements up into separate sentences. For example, in section 3.2. there are three semicolons in one sentence.
Authors response: A c-e: Major revision was performed.
Reviewers’ response: Still a lot of semicolons, but better.
- Figures need units and labels. All the figures are missing a y-axis label. The titles lack precision. There are not any units on x-axis. The general quality of the figures is poor. In addition, giving your figures a topic sentence with the take home message would help clarify what you are trying to say. For example, in figure 2 are you trying to say, all patients that exhibited a metastatic event had cellular variation in their original tumor.
Authors response: Af: Regarding reviewer comments, we have changed all graphs with better explanations and resolution.
Reviewers’ response: Better.
- In the discussion section, “eighter” should be either? Authors response: NO response.
Reviewers’ response: Still not fixed.
We corrected it, Thank you.
- The article needs a representative figure showing histological staining and classification of commonly referenced features. For example, the authors reference focal dedifferentiated nodule, cellular variant, mitotic count, etc. yet how these terms were quantified, or the definition of these terms is never mentioned in the manuscript. Adding a histological figure and defining these terms would increase the clarity of the manuscript.
Authors response: A3: Pathologic description of the dedifferentiated nodules was added to the manuscript. As we mentioned in the discussion, the hypercellular variant of the tumor is mostly a subjective parameter, and no well-defined cut-off has been determined. Besides, there
are some degrees of interobserver discrepancy to evaluate cellularity [1,2]. In our study,
the hypercellular variant and a hypercellular component were interchangeably utilized
in comparison to the hypocellular hyalinized variant. Having the histological photos was
not part of our proposal, but we can consider it in case it is necessary to explain the
study findings.
Reviewers’ response: A pathologic description of the dedifferentiated nodules was not found in the manuscript. In addition, the interobserver discrepancy and ambiguity of the terms highlight the need for a representative figure and standardization even more.
Answer: Dedifferentiation in our SFTs was defined by a high-grade pleomorphic sarcoma in an abrupt transition from typical SFT.
We added the histological figures to the result section based on your comment. You may find it in Figure 6 (line 202-207) in the revised manuscript.
- The introduction is factually incorrect at times. Currently SFT is considered a mesenchymal tumor with fibroblastic phenotype. The classification of SFT does not mean hemangiopericytoma (HPC) has been abolished. According the WHO, in 2013, after the discovery of the NAB2-STAT6 fusion in both tumor types now the classifications are considered the same cancer type. However, HPC is commonly used with the primary origin is intracranial and SFT when the origin is extra-meningeal. The sentence stating with, “using a demographic feature like the patient’s age...” is confusing and needs to be broken up into two sentences.
Authors response: A4: according to the 2021 WHO classification of the tumors of CNS, the term “hemangiopericytoma” has been retired, with the tumor now termed only Solitary fibrous tumor (rather than the hybrid term “Solitary fibrous tumor/hemangiopericytoma” used in the 2016 CNS classification) [3]
Reviewers’ response: Thank you for the reference. Reviewer stands corrected.
- Did any of the patients have metastasis at presentation?
Authors response: No, only one case was suspicious to have a lung metastasis simultaneously; however, the metastatic lesion had not been found at the time of the index surgery.
Reviewers’ response: Ok, this should be stated in the manuscript.
We have added the above-mentioned finding (section 3.1, line 109-111).
- Why was the patient with a late local recurrence excluded from the statistical analysis for metastasis? See results section 3.1.
Authors response: A6: Some authors believe that the local recurrence is mostly impacted by the surgical margin. We performed our analysis purely on distant metastases to eliminate the impact of
any technical confounder such as positive margin or radiotherapy, ... on the biological
behavior of the tumor.
Reviewers’ response: Ok. Did you mean to say you removed her from the statistical analyses for local recurrence then? May also want to mention the positive margin and your reasoning in the manuscript.
The result section 3.1 was revised to avoid any ambiguity.
- Does figure 3 include mitotic rate of metastatic tumors too?
Authors response: A7&8: Regarding reviewer comments, we have changed all graphs with better explanations and resolution.
Reviewers’ response: From the marked version, it is hard to tell what the final figures are. To question 7, the mitotic rate is often different between the primary tumor and the tumor metastases. Does the figure plot the mitotic counts of the primary tumor, tumor met, or both?
Answer: Since size and mitosis were considered continuous variables in our study and following the requested changes in the analysis, the previous graph was replaced with Kaplan Meier curve of metastatic events. Therefore, the new figure 4 (previously 3) doesn’t include mitotic rate anymore and we included size and mitotic count analysis only in table 2 and text in the manuscript.
- A line graph may help understand the correlation between a feature (number of mitoses, size, etc) and survival.
Authors response: A7&8: Regarding reviewer comments, we have changed all graphs with better explanations and resolution.
Reviewers’ response: From the marked version, it is hard to tell what the final figures are. It seems like the new figure 3 doesn’t include mitotic rate anymore. Also, the caption says probability of metastasis on the y-axis but the figure says, “survival probability (%)”. Which one is it?
Answer: As mentioned previously, the previous graph was replaced with Kaplan Meier curve of metastatic event. “Survival analysis” on the y-axis of figure 4 (previously 3) is a general term. The metastasis is considered as an event, and each incident of metastasis decreases survival. The figure caption was already changed.
- Regarding Table 2, Please briefly define OR (odd ratio), the main text and table p values do not match (DeMicco’s), why isn’t Pasquali’s score mentioned in Table 2?
Authors response: We performed Cox regression analysis and replaced the odd ratio with Hazard Ratio.
Regarding Demicco’s score, we analyzed the p-value of association between outcomes and both numerical and categorical Demicco’s scores while we only mentioned the p-value belonging to the numerical Demicco’s score in the table2. We revised the table to eliminate the vague presentation.
Reviewers’ response: Better.
- Negative surgical margins could be a significant factor contributing to metastasis. Additional analysis on those patients with negative surgical margins vs those with wide margins should be analyzed.
Authors response: A10: As we mentioned in A6, the margin directly affects the local recurrence rate, and any association between the margin and metastasis is mostly due to the increased rate of
metastases among recurrent lesions. Our study plan was to be focused on the independent
variables in the primary tumor that may predict the risk of metastasis.
Reviewers’ response: Ok. Thank you for explaining your logic for not including margins in your analysis.
- In the discussion section, other articles including meningeal SFTs when correlating cellularity is not a “drawback.” Moreover, cellularity should be quantified/defined. Lastly, please elaborate on physiologically why you think larger than 7cm tumors have a high probability of aggressive behavior?
Authors response: A11: previous studies suggested that the location may impact the behavior of SFTs [1,4]; hence, the local could be a confounder for the statistical association between cellularity and outcomes in the meningeal SFTs. The comment regarding cellularity
was already addressed in A3. In our study plan, the distant metastases were
considered as an aggressive biological behavior of the Tumor.
Reviewers’ response: The authors answer is unclear. Cellularity is not clearly defined/shown in the manuscript.
Answer: The cellular SFT (“classic hemangiopericytoma”) is characterized by a tightly packed proliferation of ovoid to spindle cells arranged around conspicuous vessels and scant stromal components. This area is commonly associated with other aggressive histologic features including large size, increased mitotic activity, cytologic atypia, or necrosis. We added the histological figures to the manuscript to resolve the ambiguity. You may find it in Figure 2 (line 139-143) in the revised manuscript.
Ok, after consulting the manuscript, the authors have corrected the discussion sentence to now say size correlates with metastasis which correlates with patient survival. Still the authors do not hypothesize on why size may increase metastases, but that is not required for this manuscript.
- Pasquali S, Gronchi A, Strauss D, Bonvalot S, Jeys L, Stacchiotti S, et al. Resectable extra-pleural and extra-meningeal solitary fibrous tumours: a multi-centre prognostic study. European Journal of Surgical Oncology (EJSO). 2016;42(7):1064-70.

Reviewer 3 Report
In the revised manuscript, authors addressed almost all the points to issue, except for the histological figure of dedifferentiated SFT component. It should be essential. Except this issue, I think that the manuscript reached the enough level for publication. Additional histopathological figures should be necessary.
Author Response
Reviewer 3, Round 2
February 3, 2023
Dear Reviewer,
We are extremely grateful for having your expert feedback on our manuscript; we appreciate it. Although we had some limitations in this retrospective study, we tried our best to answer all the questions and revised our manuscript and figures based on your insightful comments.
Comment:
In the revised manuscript, authors addressed almost all the points to issue, except for the histological figure of dedifferentiated SFT component. It should be essential. Except this issue, I think that the manuscript reached the enough level for publication. Additional histopathological figures should be necessary.
Answer:
Dedifferentiation in our SFTs was defined by a high-grade pleomorphic sarcoma in an abrupt transition from typical SFT.
We added the histological figures to the result section based on your comment. You may find it in Figure 6 (lines 202-207) in the revised manuscript.

Round 3
Reviewer 2 Report
See attached PDF.

Author Response
Dear Reviewer,
We sincerely appreciate your time and insightful comments. We tried to answer explicitly the questions and revise our manuscript and figures based on your expert opinion.
Reviewer’s response: Thank you for adding figure 1 and figure 2 showing H&E histology to this manuscript. Please follow proper figure guidelines to make the figure more professional and clearer. That is, improve the scale bar to be readable. Make the image resolution the same/similar between the two pictographs (i.e. same scale bar and image size). Include arrows to indicate what you are referring to in “upper” and “lower”. Include subfigure labels (i.e. A and B) to indicate the image you are referring to (rather than Left and Right). In addition, if these tumors had STAT6 staining, and/or a proliferation marker, adding these images below the H&E would further decrease the ambiguity of what a dedifferentiate vs. undifferentiated SFT looks like.
Author’s Response: Thank you for your valuable comment.
The photos were uniformly edited (1652*991 pixels, DPI 300 pixels/inch); moreover, high-quality photos are accessible as supplementary materials for the article. Arrows were added to indicate the classic SFT and the dedifferentiated components (figure 6A). The figure legends were also revised to resolve any ambiguity (lines 206-212). Unfortunately, there is no photo for the presented cases to show STAT6 staining features.
Reviewer’s response: The histological images help understand what is meant by cellularity of the tumor. However, it is unclear why the term “hemangiopericytoma” is being used. Which of the two tumor types has the characteristic staghorn vessels? Also, if you keep the resolution (scale) of the images the same it will be easier to compare the two. Try to make the scale bar legible. If able, include STAT6 and proliferation stains to this Figure.
Author’s Response: Thank you for your great suggestion. The term” hemangiopericytoma” was removed and the figure legend was revised (138-144); also, the photos were uniformly edited (1652*991 pixels, DPI 300 pixels/inch). All the histologic explanations and related figure legends were provided by our sarcoma pathologist (S.J). Furthermore, as mentioned above, high-quality photos are accessible as supplementary materials for the article.
